computational biology

mathematical model, multiple myeloma, GADD45β|MKK7 complex, the NF-κB pathway

**Author for correspondence:**
Bing Ji
e-mail: b.ji@sdu.edu.cn

# Mathematical modelling of the role of GADD45β in the pathogenesis of multiple myeloma

Yao Zhang[1], Changqing Zhen[2], Qing Yang[3] and Bing Ji[1]

[1]School of Control Science and Engineering, Shandong University, Jinan 250061, People's Republic of China
[2]Department of Hematology, and [3]Department of Breast and Thyroid Surgery, Shandong Provincial Hospital Affiliated to Shandong University, Jinan 250021, People's Republic of China

QY, 0000-0002-0884-7869; BJ, 0000-0003-1326-4120

Multiple myeloma (MM) is an incurable disease with relatively high morbidity and mortality rates. Great efforts were made to develop nuclear factor-kappa B (NF-κB)-targeted therapies against MM disease. However, these treatments influence MM cells as well as normal cells, inevitably causing serious side effects. Further research showed that NF-κB signalling promotes the survival of MM cells by interacting with JNK signalling through growth arrest and DNA damage-inducible beta (GADD45β), the downstream module of NF-κB signalling. The GADD45β-targeted intervention was suggested to be an effective and MM cell-specific treatment. However, the underlying mechanism through which GADD45β promotes the survival of MM cells is usually ignored in the previous models. A mathematical model of MM is built in this paper to investigate how NF-κB signalling acts along with JNK signalling through GADD45β and MKK7 to promote the survival of MM cells. The model cannot only mimic the variations in bone cells, the bone volume and MM cells with time, but it can also examine how the NF-κB pathway acts with the JNK pathway to promote the development of MM cells. In addition, the model also investigates the efficacies of GADD45β- and NF-κB-targeted treatments, suggesting that GADD45β-targeted therapy is more effective but has no apparent side effects. The simulation results match the experimental observations. It is anticipated that this model could be employed as a useful tool to initially investigate and even explore potential therapies involving the NF-κB and JNK pathways in the future.

## 1. Introduction

Multiple myeloma (MM) is a neoplastic plasma cell disease with high morbidity and mortality rates. MM reportedly accounts for

approximately 10–15% of haematologic malignancies in the world [1,2]. Almost 6–7 individuals out of every 100 000 people in the world are diagnosed with MM each year [2]. In 2015, 28 850 patients were diagnosed with MM, and the number of deaths due to this illness was more than 11 000 in the United States [3]. However, MM is still incurable and there is a great need to develop more effective therapeutic strategies.

Nuclear factor-kappa B (NF-κB) was reported to be abnormally activated in many types of cancer cells, including MM in the 1990s [4], and it inhibits the apoptosis of cancer cells by upregulating anti-apoptotic genes [4,5]. A series of studies have been performed to investigate the underlying mechanism of NF-κB in the pathogenesis of MM, with the aim of exploring NF-κB-targeted therapies [6–11]. However, to date, no appropriate NF-κB-targeted treatments have been established due to their serious side effects [4]. The inhibitors of NF-κB similarly affect the NF-κB pathway in MM and normal cells leading to serious damage. This reflects the important roles of NF-κB in cell survival, inflammatory and immune responses and other issues [12].

Growth arrest and DNA damage-inducible beta (GADD45β) forms part of the downstream module of the NF-κB pathway and is essential to the survival of MM cells [4,13]. GADD45β binds to and inhibits MAP kinase kinase 7 (MKK7) to reduce the phosphorylation of c-Jun N-terminal kinase (JNK) pathway which usually acts to ensure the normal apoptosis of cell [14]. Therefore, the overexpression of GADD45β due to an abnormally activated NF-κB pathway in MM cells can inhibit the apoptosis of MM cells by suppressing the JNK pathway [4,14,15]. Hence, GADD45β | MKK7-targeted therapeutic strategies were suggested as a potential way to kill MM cells effectively, and crucially, these may not have side effects on normal cells [4,16].

A series of mathematical models have been constructed to study MM based on experimental findings [17–23]. These models have demonstrated great potential to improve our understanding of the complicated pathogenesis of the disease. However, the underlying mechanism through which GADD45β promotes the survival of MM cells as downstream modules of the NF-κB pathway was not included. Therefore, a mathematical model of MM is built in this paper by considering the NF-κB and JNK pathways and their interaction through GADD45β and MKK7. The model cannot only mimic the variations in bone cells, the bone volume and MM cells with time, but it can also examine how the NF-κB pathway acts with the JNK pathway to promote the development of MM cells.

# 2. Model development

## 2.1. Basic structure of the model

Figure 1 describes the schematic diagram of the constructed mathematical model in the paper. It demonstrates the mechanisms of the NF-κB and JNK pathways and shows how the interaction between these two pathways promotes the survival of MM cells. It should be noted that the coupling between MM cells and bone cells (osteoclastic and osteoblastic lineages) is not included in figure 1, but it is included in the electronic supplementary material, Figure A1 in Appendix A for convenience, since these cells have been previously studied (e.g. [23]). Figure 1 is made up of two parts: part A and part B, which involve the NF-κB and JNK pathways, respectively.

Part A shows how the NF-κB pathway is activated and how it then upregulates anti-apoptotic genes in MM cells. In the unstimulated state, NF-κB heterodimers are always kept in the IκBα | NF-κB complex in the cytoplasm. Upon the stimulation of TNF, IKK is transformed from its neutral state (denoted as IKKn) into its active state (denoted as IKKa). IKKa can phosphorylate and ubiquitinate IκBα and then release free NF-κB heterodimers from the IκBα | NF-κB complex. The free NF-κB heterodimers enter the nucleus and regulate the transcription of IκBα, A20 and GADD45β [24]. In addition to IκBα, A20 also serves as an inhibitor of the NF-κB pathway, promoting the transformation of IKKa into its unactivated IKKi form [25]. As a downstream factor of NF-κB signalling, the newly produced GADD45β is expressed at a high level in the MM cells and interacts with MKK7 in the JNK pathway, promoting the survival of MM cells [4,26,27] (detailed information is included in part B). It should be noted that IκBβ, IκBγ and IκBε can also bind to NF-κB heterodimers and inhibit their nuclear localization signals [28]. Only IκBα is considered in our model, as NF-κB heterodimers are primarily sequestered by IκBα.

Part B describes how the overexpression of GADD45β acts with MKK7 and dysregulates the JNK pathway to facilitate the survival of MM cells. There are three cascades, JNK (a member of the mitogen-activated protein kinase (MAPK) family), MKK7 (a member of the MAPK kinase family) and MAPK kinase kinase (MAPKKK) in the JNK pathway [8,29,30]. The activated kinase at the upper level phosphorylates the kinase at the next level down the cascade. TNF stimulation promotes the phosphorylation of MAPKKK mediated by reactive oxygen species (ROS) [26,31,32]. MAPKKK_P

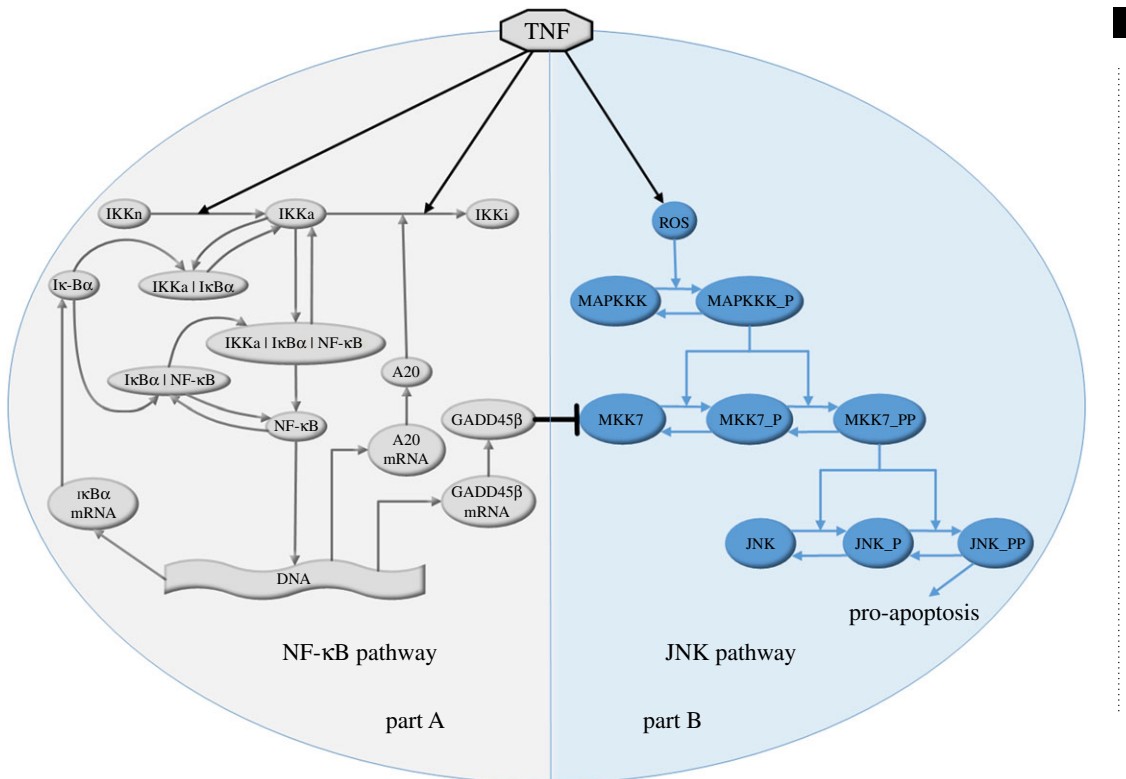

**Figure 1.** The schematic description of the mechanisms of the NF-κB and JNK pathways, and their interaction.

denotes phosphorylated MAPKKK. The full activation of both MKK7 and JNK requires the phosphorylation of two conserved sites [33]. MAPKKK_P promotes the single phosphorylation of MKK7 (denoted as MKK7_P) at first and then doubly phosphorylated (denoted as MKK7_PP). Similarly, MKK7_PP then facilitates the single and double phosphorylation of JNK (denoted as JNK_P and JNK_PP, respectively). JNK_PP is able to promote the apoptosis of MM cells by up-regulating pro-apoptotic genes [15]. The GADD45β produced during NF-κB signalling binds to MKK7 and inhibits its enzymatic activity, which limits the phosphorylation of JNK promoted by MKK7_PP. The decrease in JNK_PP inhibits the expression of pro-apoptotic genes and MM cell death.

## 2.2. Model equations

To investigate the role of GADD45β in MM pathogenesis, the mathematical model proposed in [23] is extended by including the pathways of NF-κB and JNK as well as their crosstalk. The proposed model consists of 29 ordinary differential equations (ODEs) with five state variables, namely osteoblast precursors (OBp), active osteoblasts and osteoclasts (OBa and OCa), MM and the bone volume (BV). Equations (2.1)–(2.5) represent temporal variations in the concentrations of OBp, OBa, OCa, MM and the BV, respectively. Equations (B1)–(B11) and (C1)–(C16) in the electronic supplementary material, describe variations in biochemical factors regarding the JNK and NF-κB pathways with time, respectively.

$$\frac{\mathrm{d}}{\mathrm{d}t}OB_p = D_{OB_u} \cdot \pi_{act,OB_u}^{TGF\beta} \cdot OB_u - D_{OB_p} \cdot \pi_{rep,OB_p}^{TGF\beta} \cdot \pi_{rep,OB_p}^{VCAM1} \cdot OB_p, \tag{2.1}$$

$$\frac{\mathrm{d}}{\mathrm{d}t}OB_a = D_{OB_p} \cdot \pi_{rep,OB_p}^{TGF\beta} \cdot \pi_{rep,OB_p}^{VCAM1} \cdot OB_p - A_{OB_a} \cdot \pi_{act,OB_a}^{VCAM1} \cdot OB_a, \tag{2.2}$$

$$\frac{\mathrm{d}}{\mathrm{d}t}OC_a = D_{OC_p} \cdot \pi_{act,OC_p}^{RANKL} \cdot OC_p - \pi_{act,OC_a}^{TGF\beta} \cdot A_{OC_a} \cdot OC_a, \tag{2.3}$$

$$\frac{\mathrm{d}}{\mathrm{d}t}MM = D_{MM} \cdot \pi_{act,MM}^{IL6} \cdot \pi_{act,MM}^{VCAM1} \cdot MM \cdot \left(1 - \frac{MM}{MM_{\max}}\right)$$
$$- A_{MM} \cdot \pi_{rep,MM}^{SLRPS} \cdot \pi_{act,MM}^{JNK\_PP} \cdot MM \tag{2.4}$$

and

$$\frac{\mathrm{d}}{\mathrm{d}t}BV = -K_{res} \cdot OC_a + K_{form} \cdot OB_a. \tag{2.5}$$

**Table 1.** Descriptions and values of parameters used in the model.

| parameter | description | value |
|---|---|---|
| $K_{JNK\_PP}$ | activation coefficient related to effect of JNK_PP on MM | 3.2124 pM (fitted) |
| $k_G$ | GADD45β translation rate | 0.5 s$^{-1}$ (estimated) |
| $d_G$ | GADD45β protein degradation rate | 0.0003 s$^{-1}$ (estimated) |
| $k_{GM}$ | GADD45β-MKK7 association rate | $5.3781 \times 10^{-5}$ μM s$^{-1}$ (fitted) |
| $d_{GM}$ | GADD45β-MKK7 dissociation rate | $2.1046 \times 10^{-5}$ μM s$^{-1}$ (fitted) |
| $k_{MAPKKK}$ | synthesis rate of MAPKKK | 0.5 nM s$^{-1}$ (estimated) |
| $k_{MKK7}$ | synthesis rate of MKK7 | 1.5 nM s$^{-1}$ (estimated) |
| $k_{JNK}$ | synthesis rate of JNK | 1.5 nM s$^{-1}$ (estimated) |
| $d_{MAPKKK}$ | degradation rate of MAPKKK | $2 \times 10^{-5}$ nM s$^{-1}$ (estimated) |
| $d_{MKK7}$ | degradation rate of MKK7 | $2 \times 10^{-5}$ nM s$^{-1}$ (estimated) |
| $d_{JNK}$ | Degradation rate of JNK | $2 \times 10^{-5}$ nM s$^{-1}$ (estimated) |
| $K_{D,VCAM1,MM,act}$ | half-maximal concentration of VCAM-1 on promoting the MM cells production | 0.0022 pM (fitted) |
| $K_{D,SLRPs,MM,rep}$ | half-maximal concentration of SLRPs on repressing the MM cells production | $2.3549 \times 10^9$ pM (fitted) |
| $K_{D,IL6,MM,act}$ | half-maximal concentration of IL-6 on promoting the MM cells production | $5.8217 \times 10^{-6}$ pM (fitted) |
| $MM_{max}$ | maximum possible MM concentration | 2.0836 pM (fitted) |

'Hill functions' are used to represent the cellular interaction via single ligand to receptor binding denoted by π functions [34], with equations (2.6) and (2.7) describing the stimulating and inhibiting functions of ligand-receptor binding. Here. $L$ represents the concentration of the ligand, $\beta$ represents maximal expression level of the promoter, $n$ is the coefficient which regulates the steepness of the function $\pi$ and $k_1$ and $k_2$ represent the dissociation constant, respectively. Both $\beta$ and $n$ are assumed to equal 1 in the model following the work of Pivonka et al. [34].

$$f(x) = \beta\pi_{act} = \frac{\beta(L)^n}{k_1 + (L)^n} \tag{2.6}$$

and

$$f(x) = \beta\pi_{rep} = \frac{\beta}{1 + (L/k_2)^n}. \tag{2.7}$$

$dOB_p/dt$, $dOB_a/dt$, $dOC_a/dt$, $(d/dt)MM$ and $(d/dt)BV$ denote the variations of $OB_p$, $OB_a$, $OC_a$, $MM$ and $BV$, respectively. For example, $dOB_p/dt$ is the variation of $OB_p$ with time. $D_{OB_u}$ and $D_{OB_p}$ represent the differentiation rates of uncommitted osteoblast progenitors and osteoblast precursors. $OB_u$ and $OB_p$ are concentrations of uncommitted osteoblast progenitors and osteoblast precursors. $\pi_{act,OB_u}^{TGF\beta}$ represents the stimulation of uncommitted osteoblastic progenitors into osteoblastic precursors. $\pi_{rep,OB_p}^{TGF\beta}$ represents the inhibition of the differentiation of osteoblastic precursors into active osteoblasts. $\pi_{rep,OB_p}^{VCAM1}$ represents BMSC-MM cell adhesion that blocks the differentiation of mature osteoblasts from their progenitors. Definitions of other Hill functions and variables are not repeated here but they are included in electronic supplementary material, Appendix A.

The newly added Hill function $\pi_{act,\ MM}^{JNK\_PP}$ in equation (2.4) represents the promotion of MM cell apoptosis by JNK_PP. The definition of $\pi_{act,\ MM}^{JNK\_PP}$ is as follows:

$$\pi_{act,\ MM}^{JNK\_PP} = \frac{JNK\_PP}{JNK\_PP + K_{JNK\_PP}}, \tag{2.8}$$

where $K_{JNK\_PP}$ represents the activation coefficient related to JNK_PP promoting MM cell apoptosis, and its value is included in table 1. JNK_PP represents the concentration of doubly phosphorylated JNK. The calculation of JNK_PP requires the mathematical modelling of the NF-κB and JNK pathways together

**Table 2.** The initial values of cell concentrations used in the model.

| variables | values | unit |
|-----------|--------|------|
| OBu | $3.27 \times 10^{-6}$ [36,37] | pM |
| OBp | $7.63 \times 10^{-4}$ [38] | pM |
| OBa | $6.31 \times 10^{-4}$ [39,40] | pM |
| OCp | $1.28 \times 10^{-3}$ [41] | pM |
| OCa | $1.05 \times 10^{-4}$ [39,40] | pM |
| MM | $3.26 \times 10^{-1}$ [42,43] | pM |

with their interaction through GADD45β. In accordance with the earlier work of [35] and [25], equations (B1)– (B11) and (C1)–(C16) in the electronic supplementary material, were constructed to simulate the JNK and NF-κB pathways. The distinct feature of these equations is that the interaction of the two pathways, which was ignored before, was included, with the addition of $E_{\text{MKK7}}^{\text{GADD45β}}$ in electronic supplementary material, equations (B10) and (C16). $E_{\text{MKK7}}^{\text{GADD45β}}$ represents the binding of GADD45β to MKK7, which decreases the amount of MKK7 available for phosphorylation into MKK7_P. This action further leads to the inhibition of the JNK pathway and promotes the survival of MM cells as discussed before. The definition of $E_{\text{MKK7}}^{\text{GADD45β}}$ is as follows:

$$E_{\text{MKK7}}^{\text{GADD45β}} = k_{GM} \cdot GADD45β \cdot \text{MKK7} - d_{GM} \cdot (\text{GADD45β|MKK7}) , \qquad (2.9)$$

where the first term represents the formation of the GADD45β|MKK7 complex, and the second term describes its dissociation. The definitions of the variables in the JNK and NF-κB pathways are included in electronic supplementary material, Appendices B and C. In summary, π function $\pi_{act, \ MM}^{JNK\_PP}$ and 24 ODEs are new additions to our model, allowing us to investigate the essential role of the interactions between NF-κB and JNK signalling in the development of MM.

# 3. Results

The definitions and values of the model parameters are listed in table 1. Several parameter values were reported in previous studies, while the remaining unknown parameters (i.e. those parameters where experimental data are unavailable or those that have no direct biological meaning) were fitted via the genetic algorithm (GA) in this paper. The initial values of the model variables are described in table 2. The ode45 solver in the Matlab software package (R2014b, Mathworks, Natick, USA) is used to solve the model equations. Curves in figures 2–10 represent numerical solutions of model equations. The Matlab code is included in the electronic supplementary material. The calculation of unknown model parameters based on GA are described as follows:

$$F(X) = \sum_{i=1:3} abs(M(X)_i \ - \ P_i) \qquad (3.1)$$

and

$$X = [K_{JNK_{PP}}, \ldots, MM_{\max}], \qquad (3.2)$$

where $X = [K_{JNK\_PP}, k_{GM}, d_{GM}, K_{D, VCAM1,MM,act}, K_{D, SLRPs,MM,rep}, K_{D, IL6,MM,act}, MM_{max}]$ is a row vector consisting of the seven unknown model parameters and represents one point in the parameter space. $M(X)_i$ and $P_i (i = 1, 2, 3)$ represent model outputs corresponding to each point in the parameter space and the preferred model outputs, respectively. In this work, model outputs refer to the concentrations of $OB_p$. Different combinations of parameter values have different outputs. GA is an effective way to search for parameter values in parameter space to make model outputs approach to preferred outputs.

Figures 2–4 describe the variations in the concentrations of bone cells, MM cells, the OBa : OCa ratio, and the bone volume after the invasion and removal of the MM cells. As demonstrated in figure 2, the concentrations of OBp, OBa and OCa remain in a steady state under normal conditions. This steady state is disturbed due to the invasion of the MM cells. OBp, OBa and OCa all increase to different degrees following the increase of MM cells. These simulation results are consistent with experimental observations [44–47]. OCa grows in a larger degree than that of OBa, which leads the OBa : OCa ratio

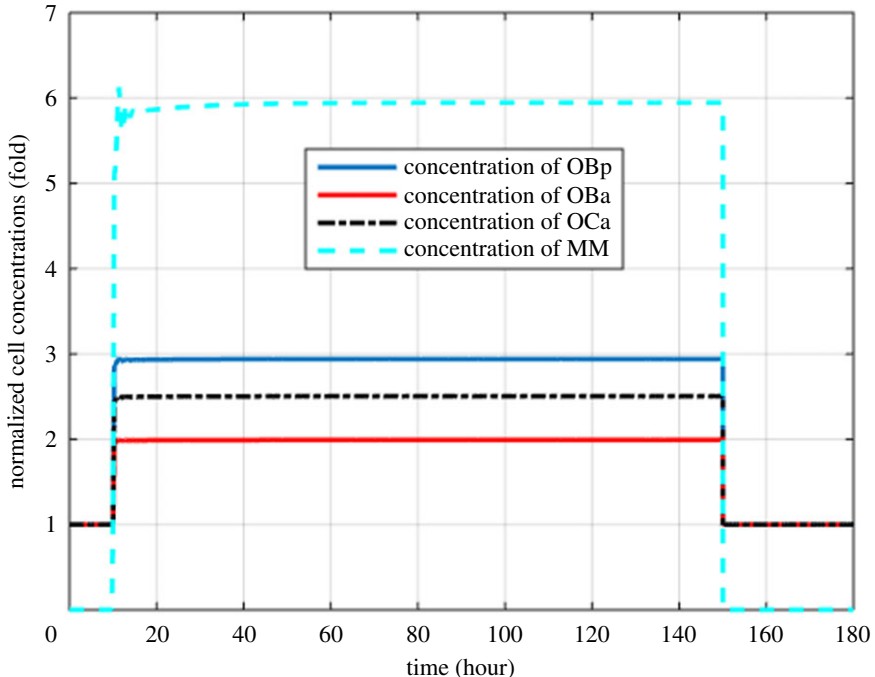

**Figure 2.** Model simulations of the normalized variation in the concentrations of OBp, OBa, OCa and MM cells with respect to their respective initial values (MM cells are injected at 10th hour and removed at 150th hour).

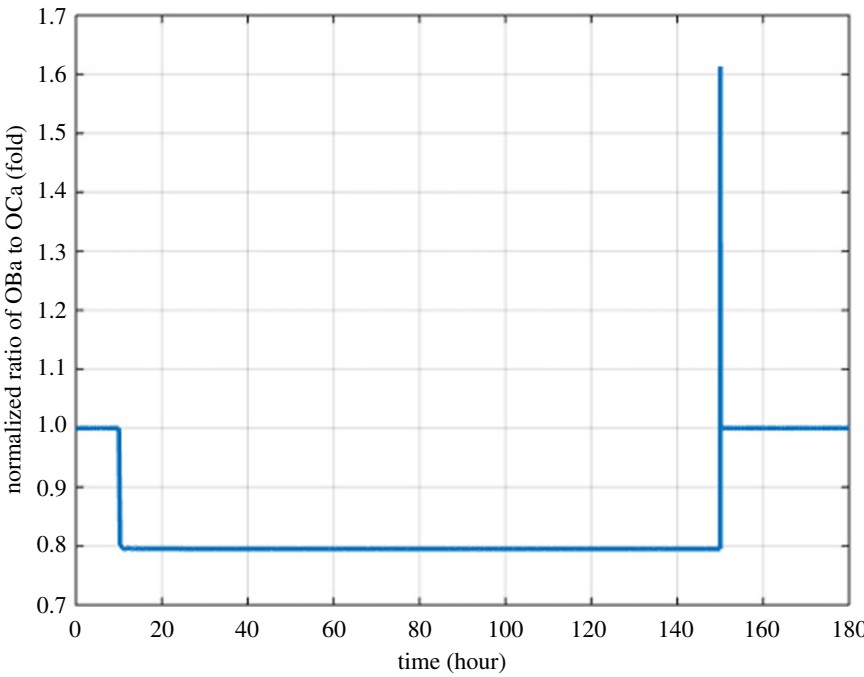

**Figure 3.** Model simulation of the variation in the normalized ratio of OBa to OCa with respect to the initial ratio (MM cells are injected at 10th hour and removed at 150th hour).

to drop after the invasion of the MM cells, as demonstrated in figure 3. The OBa : OCa ratio is a key factor in the variation of the bone volume, since OBa and OCa are for forming and resorbing bone, respectively, during the bone remodelling process. The decreased OBa : OCa ratio causes the decline in the bone volume, as shown in figure 4, which is consistent with the observed loss of the bone volume in MM patients [44]. These simulation results match the work of [23]. Figures 5 and 6 reveal the change in the GADD45β concentration after the activation of NF-κB signalling and its influence on JNK signalling. Figure 7 reveals the temporal variation in MM cells due to the inhibition of GADD45β and MKK7

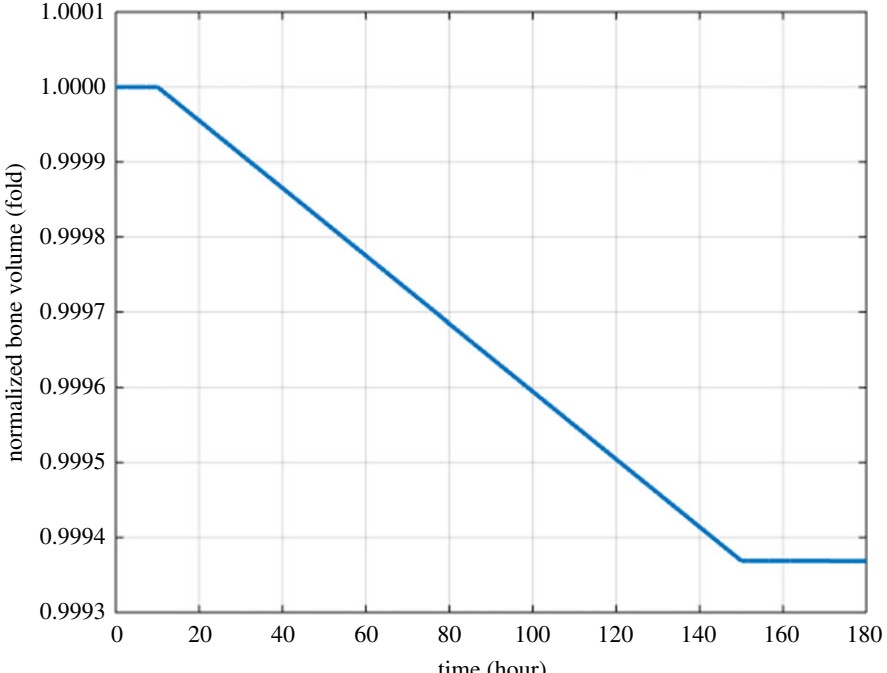

**Figure 4.** Model simulation of the variation in the normalized bone volume with respect to its initial value (MM cells are injected at 10th hour and removed at 150th hour).

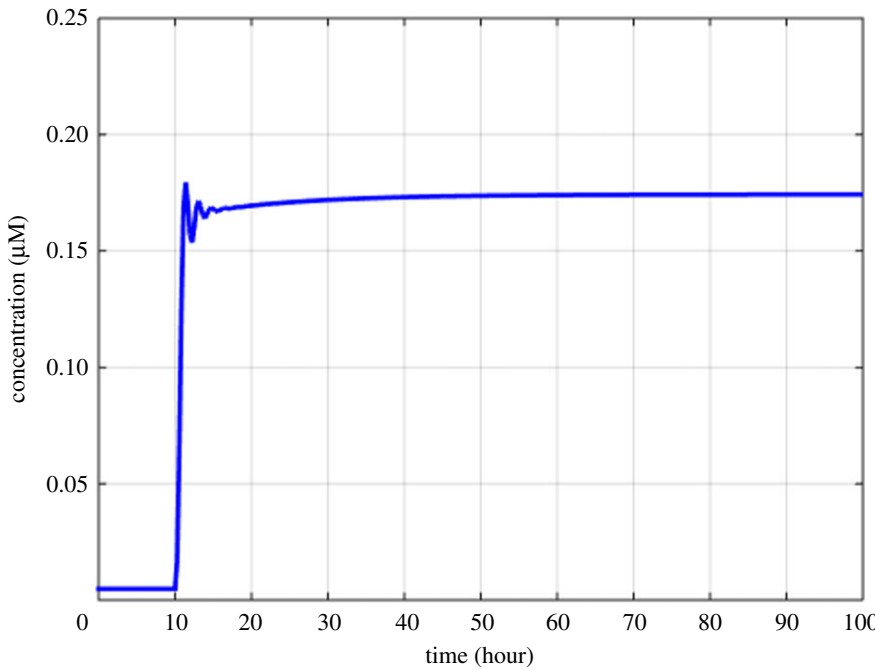

**Figure 5.** Model simulation of the variation in the GADD45β concentration after the activation of the NF-κB pathway at 10th hour.

binding by 10% and 40%. Figures 7 and 8 suggest how GADD45β-targeted therapy inhibits the growth of MM cells and JNK_PP. Thus, the inhibition of GADD45β binding to MKK7 leads to an obvious drop of MM cells, which then leads to a new steady state at a lower level. The larger degree of inhibition results in a larger drop in MM cells. These simulation results can be confirmed by the experimental data [4]. Figure 9 shows the variation in MM cells after IKK activation is inhibited by 10% and 40%. As shown in figure 9, IKK inhibition leads to a decrease in MM cells which is accentuated with a larger degree of IKK is inhibition. Figure 10 indicates how the inhibition of GADD45β and IKK influence NF-κB.

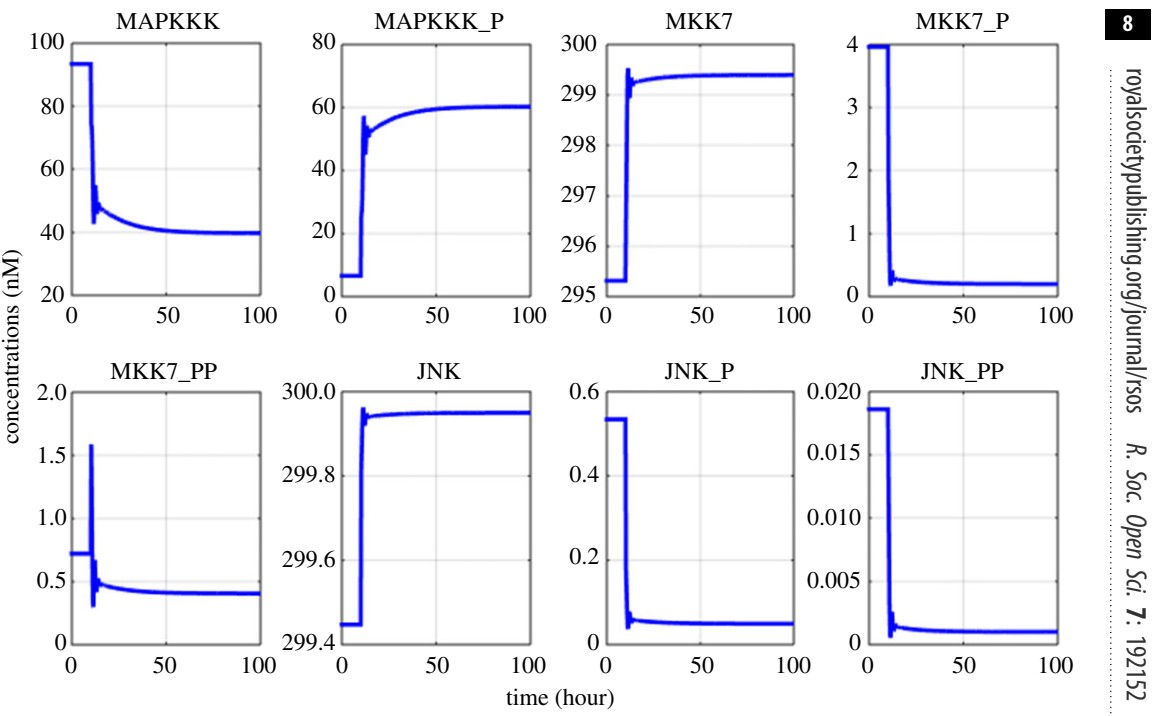

**Figure 6.** Model simulations of the variation in biochemical factors involved in JNK signalling after the pathway is activated at 10th hour.

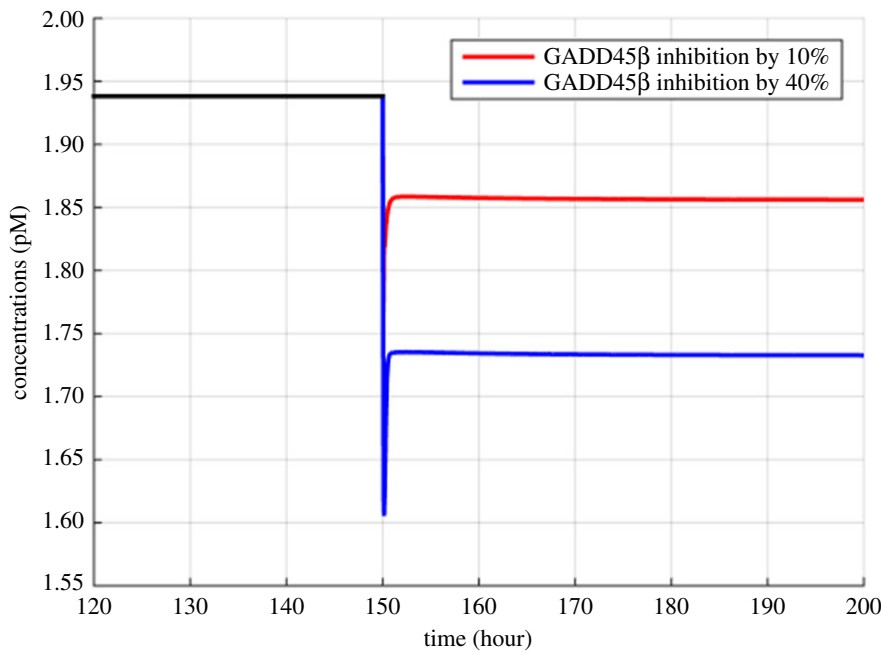

**Figure 7.** Model simulations of the variation in MM cells due to the inhibition of GADD45β and MKK7 binding by 10% and 40% at 150th hour.

## 4. Discussion

The proposed model in this paper aims to investigate how the interaction between NF-κB and JNK signalling via the GADD45β|MKK7 complex promotes the development of MM. The model can not only simulate the temporal variation of bone cells, the bone volume and biochemical factors involved in the NF-κB and JNK pathways, but it can also mimic the underlying mechanism in which the coupling between the two pathways inhibits apoptosis in MM cells. Additionally, the efficacies of GADD45β- and

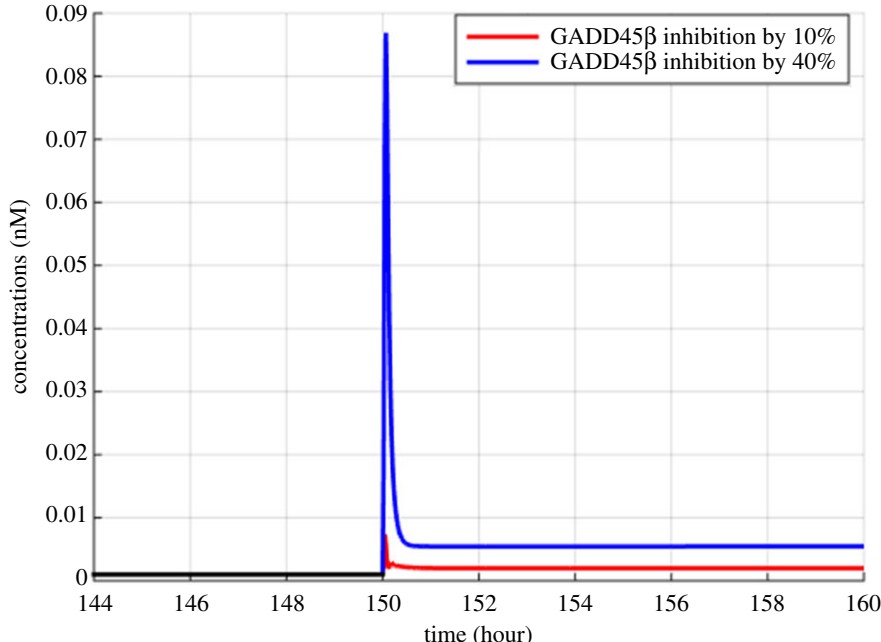

**Figure 8.** Model simulations of the variation in JNK_PP due to the inhibition of GADD45β and MKK7 binding by 10% and 40% at 150th hour.

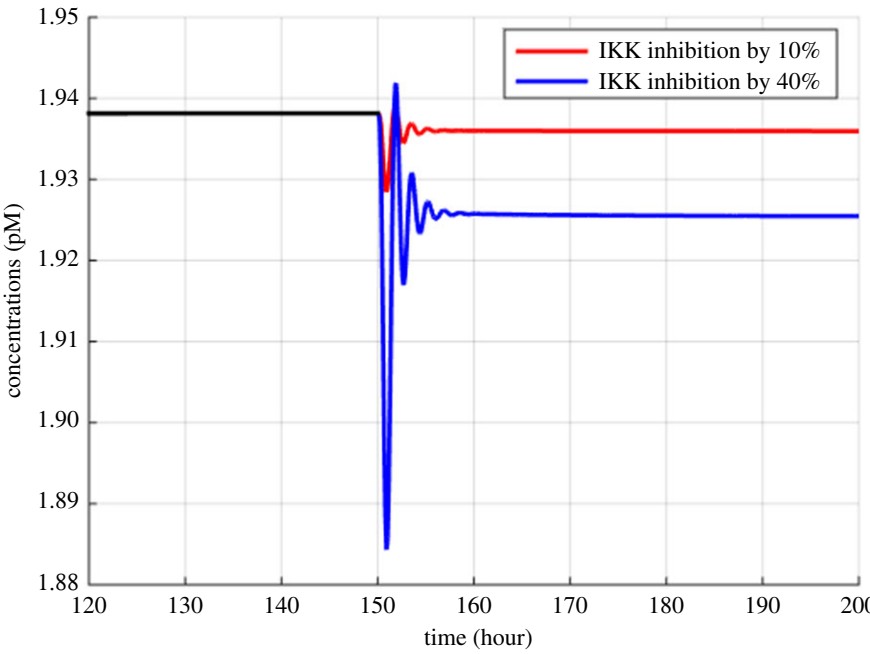

**Figure 9.** Model simulations of the variation in MM cells due to the inhibition of IKK activation by 10% and 40% at 150th hour.

NF-κB-targeted therapies were compared based on the model simulation. This work only demonstrates the temporal variation in the GADD45β concentration after the activation of the NF-κB pathway. The variations in other biochemical factors related to NF-κB signalling were not repeated here as they have been analysed in [25] but are included in electronic supplementary material, Appendix C.

As mentioned above, JNK_PP plays an essential role in the apoptosis of MM cells, and JNK_PP is able to inhibit the growth of MM cells. According to figure 8, the decrease in MM cells can be explained by the rising amount of JNK_PP after the inhibition of GADD45β binding to MKK7. NF-κB-targeted therapies are most often performed by inhibiting IKK, which is essential for NF-κB activation [8]. Based on model simulations (figure 10), the GADD45β-targeted therapy is suggested to be more sensitive than the NF-κB-targeted therapy as it can produce a greater suppression of MM cells than IKK inhibition when each are

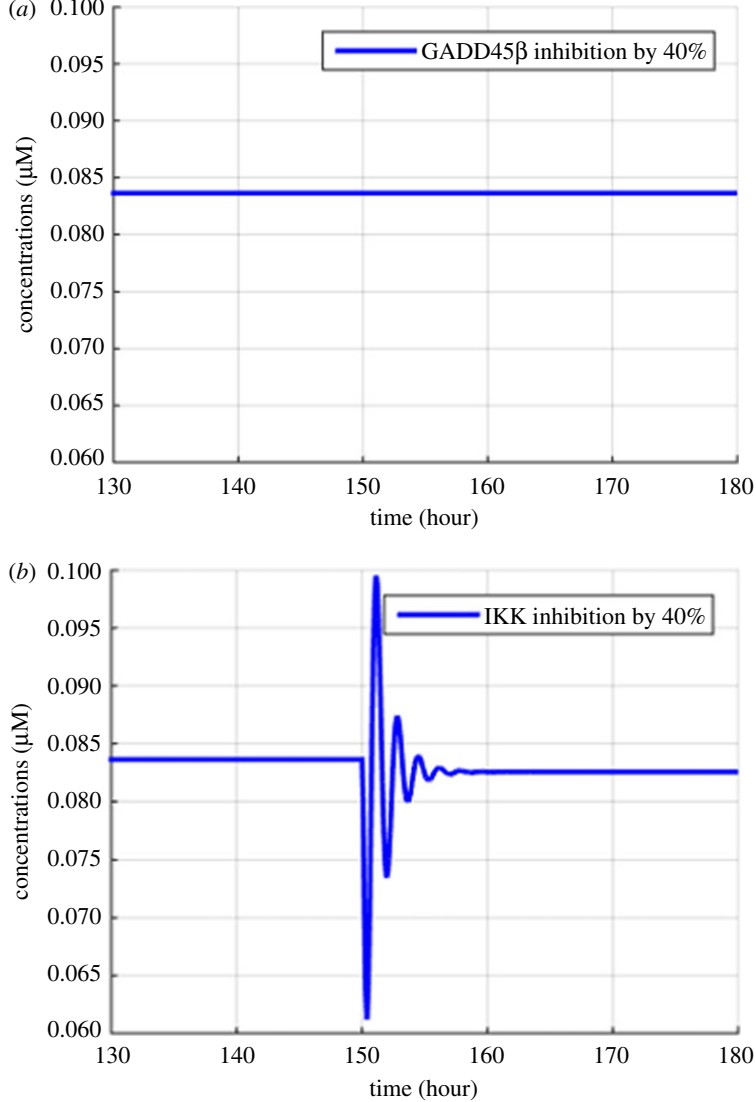

**Figure 10.** Model simulation of the variation in NF-κB due to the inhibition of binding of GADD45β and MKK7, and the IKK activation. (*a*) The binding of GADD45β and MKK7 inhibited by 40% at 150th hour. (*b*) The IKK activation inhibited by 40% at 150th hour.

suppressed to the same degree. In addition to the higher sensitivity, GADD45β-targeted therapy has a high MM cell specificity. This approach can not only kill MM cells effectively, but it also has no effect on NF-κB levels. Therefore, it can be predicted that the side effects seen with NF-κB-targeted treatment would be avoided. However, these *in silico* hypotheses generated from the mathematical model described in this paper will ultimately have to be validated by further experimental data to confirm the clinical potential of GADD45β-targeting therapies. It should also be noted that the proposed simplified model does not take into account the interactions with several other proteins and other cell structures, which represents a limitation of this study. Specifically, although the differentiation of progenitors into active osteoclasts and osteoblasts contains several intermediate stages, the model only considered four osteoblastic and three osteoclastic lineages, and contained three state variables: osteoblast precursors, active osteoblasts and active osteoclasts. Additionally, IκBs, including IκBα, IκBβ, IκBγ and IκBε, can also bind to NF-κB and form a complex but we only considered IκBα.

# 5. Conclusion

This paper describes a mathematical model of MM that was used to investigate how NF-κB signalling acts to include the interplay between NF-κB and JNK signalling that was not included in the previous models.

The model not only reconstructs how the invasion of MM cells disturbs the steady state of the bone microenvironment and triggers the variation in bone cells, but it also mimics the changes in biochemical factors involved in the NF-κB and JNK pathways. In addition, the model also investigates the efficacies of GADD45β- and NF-κB-targeted treatments, suggesting that GADD45β-targeted therapy is more effective but has no apparent side effects. The simulation results match the experimental observations. This model helps to illuminate the essential function of the crosstalk between the NF-κB and JNK pathways during MM development. It is anticipated that this model could be employed as a useful tool to initially investigate and even explore potential therapies involving the NF-κB and JNK pathways. More work is required to improve the built model to take into account many others parameters such as the interactions with other protein partners, with membranes and other cell structures.

Data accessibility. Matlab code is available in the electronic supplementary material.

Authors' contributions. Y.Z. carried out program coding and model simulations, participated in the design of the study and drafted the manuscript; C.Z. carried out the analysis of experimental data and critically revised the manuscript; Q.Y. collected the experimental data and critically revised the manuscript; B.J. conceived of the study, designed the study, coordinated the study and helped draft the manuscript. All authors gave final approval for publication and agree to be held accountable for the work performed therein.

Competing interests. We declare that we have no competing interests.

Funding. This work was supported by National Key R&D Program of China (grant no. 2018YFB1305400); the National Natural Science Foundation of China (grant nos. 61673246 and 81301294); the key Research and Development Program of Shandong province (grant no. 2016GSF201168) and the Research and Development Program of Jinan (grant no. 201907064).

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
