## [Reviewer comments · Royal Society Open Science]

Review History

RSOS-192152.R0 (Original submission)

Review form: Reviewer 1

Is the manuscript scientifically sound in its present form?

No

Are the interpretations and conclusions justified by the results?

Yes

Is the language acceptable?

Yes

Do you have any ethical concerns with this paper?

No

Have you any concerns about statistical analyses in this paper?

No

Recommendation?

Major revision is needed (please make suggestions in comments)

Comments to the Author(s)

The authors present an extension of a previous mathematical model developed to study multiple myeloma bone disease. In this extension, the authors explore the interplay between NF- κ B and JNK signalling pathways in the context of promotion of multiple myeloma cell survival. Interestingly, the numerical simulations show a potential therapeutic opportunity to further explore in relation to GADD45 β -targeted therapy.

The numerical simulations are interesting from the biological standpoint. Several equations and parameter values are retrieved from the literature, citing relevant works. However, my biggest concern is that there is no explicit mention to the methods employed to find the parameter values introduced in this work. In the spite of the journal's effort of publishing scientifically strong and meaningful contributions to the literature, I would strongly encourage the authors to describe the methodology for choosing *the* set of parameters used in this manuscript.

Also, I would invite the authors to mention the drawbacks and limitations of the mathematical model. Models that describe biological processes related to clinical expectations may shed some light on some key elements of the different players. But certainly, there must be some scenarios that are not suitably described by the model. Perhaps the authors may mention how the limitations of the current mathematical model may be surmounted in the future.

Review form: Reviewer 2

Is the manuscript scientifically sound in its present form?

Yes

Are the interpretations and conclusions justified by the results?

Yes

Is the language acceptable?

Yes

Do you have any ethical concerns with this paper?

No

Have you any concerns about statistical analyses in this paper?

No

Recommendation?

Accept with minor revision (please list in comments)

Comments to the Author(s)

The manuscript by Zhang et al. reports a mathematical model which aims to describe how NF- κ B signalling acts in conjunction with JNK activation via GADD45 β /MKK7 to promote multiple myeloma development. In addition, the model simulates how multiple myeloma cells affect bone marrow microenvironment and attempts to describe the efficacy of GADD45 β -targeting therapies.

The overall in-silico approach is interesting and could provide a relevant tool for analysing experimental findings and eventually predicting tumour behaviour, thus complementing the current molecular understanding of Multiple Myeloma pathogenesis. I do not have any major points and I find that this manuscript provides an interesting mathematical compendium to the existing experimental data in multiple myeloma.

Minor points:

1. It would be important to mention in the discussion that any in-silico hypothesis generated from the mathematical model described in this paper will ultimately have to be validated by further experimental data to confirm the clinical potential of GADD45 β -targeting therapies.

2. The information currently in the "discussion" explaining what is demonstrated by each figure should be moved in the more appropriate "results" section.

3. The first part of the "conclusions" is repetitive and should be trimmed out considerably to avoid overlap with the introduction.

4. Page 4, Line 21

The following reference should be added at the end of the sentence "but they do not have toxic side effects on normal cells [4]": Tornatore L et al. Preclinical toxicology and safety pharmacology of the first-in-class GADD45 β /MKK7 inhibitor and clinical candidate, DTP3. *Toxicol Rep.* 2019;6:369–379. Published 2019 Apr 19. doi:10.1016/j.toxrep.2019.04.006.

2. Page 4, Line 21

The following reference should be added at the end of the sentence "but they do not have toxic side effects on normal cells [4]":

Tornatore L et al. Preclinical toxicology and safety pharmacology of the first-in-class GADD45 β /MKK7 inhibitor and clinical candidate, DTP3. *Toxicol Rep.* 2019;6:369–379. Published 2019 Apr 19. doi:10.1016/j.toxrep.2019.04.006.

Review form: Reviewer 3

Is the manuscript scientifically sound in its present form?

No

Are the interpretations and conclusions justified by the results?

Yes

Is the language acceptable?

Yes

Do you have any ethical concerns with this paper?

No

Have you any concerns about statistical analyses in this paper?

Yes

Recommendation?

Major revision is needed (please make suggestions in comments)

Comments to the Author(s)

The author claim an algorithm that can help to shed light on the complex mechanisms underlying the interplay between the growth of cancer cells (Multiple Myeloma, MM) and several molecular players known to have a role in the development of this disease. In particular they propose very complex mathematical equations/models that should describe the changes in concentration of the MM cells, and of Gadd45b, MKK7 (which interact each other), phosphorylated MKK7 (P-MKK7), doubly phosphorylated MKK7 (MKK7-PP), phosphorylated JNK (JNK-P), doubly phosphorylated JNK (JNK-PP) and various forms of IKK. The work strongly suffers of the lack of

a basic explanation of the mathematical models which prevents the understanding by non-specialists and even by specialists in the field of the MM/Gadd45/MKK/JNK biology.

In particular the Methods are only superficially mentioned and a simplified description of the model is strongly needed. Also the results is a simple enunciation of the of the figure features, while no explanation is given of how the curves are generated.

A warning must be clearly given in the discussion and conclusion sections to underline that the model is simplified and does not take into account many others parameters such as the interactions with other protein partners, with membranes and other cell structures.

The manuscript is of potential interest for the readership of the journal after providing the basic instructions for its comprehension. Major revisions are needed.

Decision letter (RSOS-192152.R0)

17-Mar-2020

Dear Dr Ji,

The editors assigned to your paper ("Mathematical modelling of the role of GADD45 β in the pathogenesis of multiple myeloma") have now received comments from reviewers. We would like you to revise your paper in accordance with the referee and Associate Editor suggestions which can be found below (not including confidential reports to the Editor). Please note this decision does not guarantee eventual acceptance.

Please submit a copy of your revised paper before 09-Apr-2020. Please note that the revision deadline will expire at 00.00am on this date. If we do not hear from you within this time then it will be assumed that the paper has been withdrawn. In exceptional circumstances, extensions may be possible if agreed with the Editorial Office in advance. We do not allow multiple rounds of revision so we urge you to make every effort to fully address all of the comments at this stage. If deemed necessary by the Editors, your manuscript will be sent back to one or more of the original reviewers for assessment. If the original reviewers are not available, we may invite new reviewers.

- Data accessibility

If you wish to submit your supporting data or code to Dryad (<http://datadryad.org/>), or modify your current submission to dryad, please use the following link:
<http://datadryad.org/submit?journalID=RSOS&manu=RSOS-192152>

- Competing interests

- Authors' contributions

- Acknowledgements

- Funding statement

Kind regards,

Andrew Dunn

on behalf of Prof Mark Chaplain (Subject Editor)

Associate Editor's comments:

Thank you for submitting to Royal Society Open Science. Upon submitting your revised manuscript, please ensure that you address all comments provided by all referees within a point-by-point response. Specifically, please ensure that any clarifications or descriptions are provided, and that the limitations of your study are properly addressed.

Comments to Author:

Reviewers' Comments to Author:

Reviewer: 1

Comments to the Author(s)

The authors present an extension of a previous mathematical model developed to study multiple myeloma bone disease. In this extension, the authors explore the interplay between NF- κ B and JNK signalling pathways in the context of promotion of multiple myeloma cell survival. Interestingly, the numerical simulations show a potential therapeutic opportunity to further explore in relation to GADD45 β -targeted therapy.

The numerical simulations are interesting from the biological standpoint. Several equations and parameter values are retrieved from the literature, citing relevant works. However, my biggest concern is that there is no explicit mention to the methods employed to find the parameter values introduced in this work. In the spite of the journal's effort of publishing scientifically strong and meaningful contributions to the literature, I would strongly encourage the authors to describe the methodology for choosing *the* set of parameters used in this manuscript.

Also, I would invite the authors to mention the drawbacks and limitations of the mathematical model. Models that describe biological processes related to clinical expectations may shed some light on some key elements of the different players. But certainly, there must be some scenarios that are not suitably described by the model. Perhaps the authors may mention how the limitations of the current mathematical model may be surmounted in the future.

Reviewer: 2

Comments to the Author(s)

The manuscript by Zhang et al. reports a mathematical model which aims to describe how NF- κ B signalling acts in conjunction with JNK activation via GADD45 β /MKK7 to promote multiple myeloma development. In addition, the model simulates how multiple myeloma cells affect bone marrow microenvironment and attempts to describe the efficacy of GADD45 β -targeting therapies.

The overall in-silico approach is interesting and could provide a relevant tool for analysing experimental findings and eventually predicting tumour behaviour, thus complementing the current molecular understanding of Multiple Myeloma pathogenesis. I do not have any major points and I find that this manuscript provides an interesting mathematical compendium to the existing experimental data in multiple myeloma.

Minor points:

1. It would be important to mention in the discussion that any in-silico hypothesis generated from the mathematical model described in this paper will ultimately have to be validated by further experimental data to confirm the clinical potential of GADD45 β -targeting therapies.

2. The information currently in the "discussion" explaining what is demonstrated by each figure should be moved in the more appropriate "results" section.

3. The first part of the "conclusions" is repetitive and should be trimmed out considerably to avoid overlap with the introduction.

4. Page 4, Line 21

The following reference should be added at the end of the sentence "but they do not have toxic side effects on normal cells [4]": Tornatore L et al. Preclinical toxicology and safety pharmacology of the first-in-class GADD45 β /MKK7 inhibitor and clinical candidate, DTP3. *Toxicol Rep.* 2019;6:369-379. Published 2019 Apr 19. doi:10.1016/j.toxrep.2019.04.006.

2. Page 4, Line 21

The following reference should be added at the end of the sentence "but they do not have toxic side effects on normal cells [4]":

Tornatore L et al. Preclinical toxicology and safety pharmacology of the first-in-class GADD45 β /MKK7 inhibitor and clinical candidate, DTP3. *Toxicol Rep.* 2019;6:369-379. Published 2019 Apr 19. doi:10.1016/j.toxrep.2019.04.006.

Reviewer: 3

Comments to the Author(s)

The author claim an algorithm that can help to shed light on the complex mechanisms underlying the interplay between the growth of cancer cells (Multiple Myeloma, MM) and several molecular players known to have a role in the development of this disease. In particular they propose very complex mathematical equations/models that should describe the changes in concentration of the MM cells, and of Gadd45b, MKK7 (which interact each other), phosphorylated MKK7 (P-MKK7), doubly phosphorylated MKK7 (MKK7-PP), phosphorylated JNK (JNK-P), doubly phosphorylated JNK (JNK-PP) and various forms of IKK. The work strongly suffers of the lack of a basic explanation of the mathematical models which prevents the understanding by non-specialists and even by specialists in the field of the MM/Gadd45/MKK/JNK biology.

In particular the Methods are only superficially mentioned and a simplified description of the model is strongly needed. Also the results is a simple enunciation of the of the figure features, while no explanation is given of how the curves are generated.

A warning must be clearly given in the discussion and conclusion sections to underline that the model is simplified and does not take into account many others parameters such as the interactions with other protein partners, with membranes and other cell structures.

The manuscript is of potential interest for the readership of the journal after providing the basic instructions for its comprehension. Major revisions are needed.

Author's Response to Decision Letter for (RSOS-192152.R0)

See Appendix A.

RSOS-192152.R1 (Revision)

Review form: Reviewer 1

Is the manuscript scientifically sound in its present form?

Yes

Are the interpretations and conclusions justified by the results?

Yes

Is the language acceptable?

Yes

Do you have any ethical concerns with this paper?

No

Have you any concerns about statistical analyses in this paper?

No

Recommendation?

Accept as is

Comments to the Author(s)

My previous concerns about the manuscript were satisfactorily addressed by the authors. I acknowledged the authors' efforts put towards addressing my fellow reviewers important comments. This is an interesting paper with model-driven hypotheses that, hopefully, could guide biologists in experiments to validate them and gain further insight about the regulatory network.

Review form: Reviewer 3

Is the manuscript scientifically sound in its present form?

Yes

Are the interpretations and conclusions justified by the results?

Yes

Is the language acceptable?

Yes

Do you have any ethical concerns with this paper?

No

Have you any concerns about statistical analyses in this paper?

Yes

Recommendation?

Accept as is

Comments to the Author(s)

Accept

Decision letter (RSOS-192152.R1)

15-Apr-2020

Dear Dr Ji,

It is a pleasure to accept your manuscript entitled "Mathematical modelling of the role of GADD45 β in the pathogenesis of multiple myeloma" in its current form for publication in Royal Society Open Science. The comments of the reviewer(s) who reviewed your manuscript are included at the foot of this letter.

Please ensure that you send to the editorial office an editable version of your accepted manuscript. Failure to provide this file may delay the processing of your proof.

on behalf of Prof Mark Chaplain (Subject Editor)
openscience@royalsociety.org

Reviewer comments to Author:
Reviewer: 1

Comments to the Author(s)
My previous concerns about the manuscript were satisfactorily addressed by the authors. I acknowledged the authors' efforts put towards addressing my fellow reviewers important comments. This is an interesting paper with model-driven hypotheses that, hopefully, could guide biologists in experiments to validate them and gain further insight about the regulatory network.

Reviewer: 3

Comments to the Author(s)
Accept

Appendix A

Response to Reviewers' comments

We are particularly grateful to Reviewers who must have spent considerable time on reviewing our work. His/her comments are especially helpful and constructive, and we feel have led to significant improvements in the paper.

Our responses below are in blue. Additions and modifications to the paper are in red.

Reviewer#1

Comments to the Author(s)

The authors present an extension of a previous mathematical model developed to study multiple myeloma bone disease. In this extension, the authors explore the interplay between NF- κ B and JNK signalling pathways in the context of promotion of multiple myeloma cell survival. Interestingly, the numerical simulations show a potential therapeutic opportunity to further explore in relation to GADD45 β -targeted therapy.

Thanks for your comments.

1. The numerical simulations are interesting from the biological standpoint. Several equations and parameter values are retrieved from the literature, citing relevant works. However, my biggest concern is that there is no explicit mention to the methods employed to find the parameter values introduced in this work. In the spite of the journal's effort of publishing scientifically strong and meaningful contributions to the literature, I would strongly encourage the authors to describe the methodology for choosing *the* set of parameters used in this manuscript.

Author response: Agreed and thanks for your suggestions. We apologise for the unclear description of the method used in the manuscript. Several parameter values were reported in previous studies, while the remaining unknown parameters (i.e. those parameters where experimental data are unavailable or those that have no direct biological meaning) were fitted via the Genetic Algorithm (GA) in this paper.

Author action: We updated the manuscript by adding some sentences in the manuscript. Please see page 9, line 17 to 20, page 10, line 4 to 17.

2. Also, I would invite the authors to mention the drawbacks and limitations of the mathematical model. Models that describe biological processes related to clinical expectations may shed some light on some key elements of the different players. But certainly, there must be some scenarios that are not suitably described by the model. Perhaps the authors may mention how the limitations of the current mathematical model may be surmounted in the future.

Author response: Agreed and thanks for your suggestion. The model in this paper is simplified and does not take into account the interactions with many other proteins

and other cell structures. The model will be continuously improved in the future by including more biochemical factors.

Author action: We updated the manuscript by adding some sentences in the conclusion part. Please see page 13, line 4 to 12, and page 14, line 5 to 7.

Reviewer#2

Comments to the Author(s)

The manuscript by Zhang et al. reports a mathematical model which aims to describe how NF- κ B signalling acts in conjunction with JNK activation via GADD45 β /MKK7 to promote multiple myeloma development. In addition, the model simulates how multiple myeloma cells affect bone marrow microenvironment and attempts to describe the efficacy of GADD45 β -targeting therapies.

The overall in-silico approach is interesting and could provide a relevant tool for analysing experimental findings and eventually predicting tumour behaviour, thus complementing the current molecular understanding of Multiple Myeloma pathogenesis. I do not have any major points and I find that this manuscript provides an interesting mathematical compendium to the existing experimental data in multiple myeloma.

Thanks a lot for your comments.

Minor points:

1. It would be important to mention in the discussion that any in-silico hypothesis generated from the mathematical model described in this paper will ultimately have to be validated by further experimental data to confirm the clinical potential of GADD45 β -targeting therapies.

Author response: Agree and thanks for your suggestions.

Author action: We updated the manuscript by adding some sentences in the discussion part. Please see page 13, line 2 to 4.

2. The information currently in the "discussion" explaining what is demonstrated by each figure should be moved in the more appropriate "results" section.

Author response: Agreed and thanks for your suggestion. Most sentences that demonstrate the figures in the "discussion" before have been moved to "results" section already.

Author action: We updated the manuscript by moving most sentences that demonstrate the figures in the discussion section before to results section. Please see page 10, line 20 to 21, page 11, line 1 to 19, and page 12, line 2 to 15.

3. The first part of the “conclusions” is repetitive and should be trimmed out considerably to avoid overlap with the introduction.

Author response: Agreed and thanks for your suggestion. Some sentences were deleted in the first part of the conclusion following your suggestions.

Author action: We updated the manuscript by deleting some sentences in the conclusions part. The details are in the conclusion part. Please see page 13, line 15 to 16.

4. Page 4, Line 21

The following reference should be added at the end of the sentence "but they do not have toxic side effects on normal cells [4]": Tornatore L et al. Preclinical toxicology and safety pharmacology of the first-in-class GADD45 β /MKK7 inhibitor and clinical candidate, DTP3. Toxicol Rep. 2019;6:369–379. Published 2019 Apr 19. doi:10.1016/j.toxrep.2019.04.006.

Author response: Thanks for your suggestion. The reference has been added in the manuscript.

Author action: We updated the manuscript by adding the reference in the manuscript. Please see page 4, line 5.

Reviewer#3

Comments to the Author(s)

1. The author claim an algorithm that can help to shed light on the complex mechanisms underlying the interplay between the growth of cancer cells (Multiple Myeloma, MM) and several molecular players known to have a role in the development of this disease. In particular they propose very complex mathematical equations/models that should describe the changes in concentration of the MM cells, and of Gadd45b, MKK7 (which interact each other), phosphorylated MKK7 (P-MKK7), doubly phosphorylated MKK7 (MKK7-PP), phosphorylated JNK (JNK-P), doubly phosphorylated JNK (JNK-PP) and various forms of IKK. The work strongly suffers of the lack of a basic explanation of the mathematical models which prevents the understanding by non-specialists and even by specialists in the field of the MM/Gadd45/MKK/JNK biology. In particular the Methods are only superficially mentioned and a simplified description of the model is strongly needed.

Author response: Agree and thanks for your suggestion. Some sentences that describe the model have been added in the manuscript.

Author action: We updated the manuscript by adding some sentences in the

manuscript. Please see page 7, line 11 to 21, and page 8, line 1 to 10.

2. Also the results is a simple enunciation of the of the figure features, while no explanation is given of how the curves are generated.

Author response: Thanks for your suggestion. More explanations are added to describe the figure features following your suggestions.

Following the work in references [1-3], relevant ordinary differential equations (ODEs) were established to describe the biological reaction process. Based on the ODEs mentioned above, the ode45 solver in the Matlab software package (R2014b, Mathworks, Natick, USA) is used to solve the model equations. Curves in the figures represent solutions of model equations graphically. Figure 2 to 10 were generated by this way. More details are included in the manuscript

Author action: We updated the manuscript by adding some sentences in the manuscript. Please see page 9, line 17 to 20, page 10, line 3 to 4, and line 20 to 21, and page 11, line 1 to 20.

3. A warning must be clearly given in the discussion and conclusion sections to underline that the model is simplified and does not take into account many others parameters such as the interactions with other protein partners, with membranes and other cell structures.

The manuscript is of potential interest for the readership of the journal after providing the basic instructions for its comprehension. Major revisions are needed.

Author response: Agree and thanks for your suggestion. Some sentences are added following your comments.

Author action: We updated the manuscript by adding some sentences in the manuscript. Please see page 13, line 4 to 12, and page 14, line 5 to 7.

[1] B. Ji, P.G. Genever, R.J. Patton, M.J. Fagan, Mathematical modelling of the pathogenesis of multiple myeloma-induced bone disease, *Int. j. Numer. Method. Biomed. Eng.* 30 (2014) 1085–1102.

[2] T. Lipniacki, P. Paszek, A.R. Brasier, B. Luxon, M. Kimmel, Mathematical model of NF- κ B regulatory module, *J. Theor. Biol.* 228 (2004) 195–215.

[3] B.N. Kholodenko, Negative feedback and ultrasensitivity can bring about oscillations in the mitogen-activated protein kinase cascades, *Eur. J. Biochem.* 267 (2000) 1583–1588.